# A Study on the Characteristics of Children’s Natural Activities in the Neighborhood and Their Influencing Factors: Evidence from Hangzhou, China

**DOI:** 10.3390/ijerph192316087

**Published:** 2022-12-01

**Authors:** Rui Ji, Sheng Li, Yuhan Shao

**Affiliations:** 1Department of Landscape Architecture, College of Landscape Architecture, Zhejiang Agriculture and Forestry University, Hangzhou 311300, China; 2Department of Landscape Architecture, College of Architecture and Urban Planning, Tongji University, Shanghai 200092, China

**Keywords:** children’s natural activities in the neighborhood, environmental education, China, child-friendly cities

## Abstract

In the process of urbanization, children are becoming increasingly estranged from nature. The phenomenon has received widespread attention in developed countries because of its many negative effects on children’s health and pro-nature behaviors. However, the existing studies lack the exploration of this issue in developing countries, especially with regard to China. In order to understand the characteristics of neighborhood natural activities and their influencing factors among children in China, we conducted a systematic questionnaire survey of 900 children (10–12 years old from Hangzhou City, Zhejiang Province, Eastern China) about their natural activities in the neighborhood. Children were asked to provide basic information on their neighborhood natural activities situation (frequency, duration and location), nature connectedness and environmental knowledge. The results showed that the frequency of children’s neighborhood natural activities in rural areas was less than that of children in urban areas, but the naturalness of the neighborhood natural activity sites of rural children was higher than that of urban children. Boys spent more time engaging in neighborhood natural activities than girls. Only children outlasted non-only children in terms of the duration of neighborhood natural activities. Regarding Influencing Factors, children’s nature connectedness and environmental knowledge significantly and positively predicted the frequency and location of children’s natural activities in the neighborhood, which indicates a new approach to the promotion of children’s neighborhood natural activities. Overall, how to encourage children to engage in natural activities in the neighborhood is a complex issue that needs to be studied in more regions and with more samples.

## 1. Introduction

In the context of rapid urbanization, children in many regions of the world are becoming disconnected from nature [1,2,3]. Similar to the concept of “extinction of experience ” [4] and “Disappearance nearby” [5], the American author Richard Louv describes this phenomenon as “children’s nature-deficit disorder” [6]. Specifically, the reduction of children’s activities in contact with nature leads to a decrease in children’s emotional connection with nature, cognitive level and sense of protection [7,8,9]. It also increases the risk of a series of health problems such as obesity, depression and cognitive and emotional disorders [10,11,12,13].

Based on the large number of negative effects of children’s nature-deficit disorder, children’s natural activities in the neighborhood, which are the most common way for children to directly experience nature in their daily lives, had attracted the attention of a large number of researchers. The relevant topics, which are currently covered in three main parts, are as follows:

### 1.1. Assessment of Children’s Natural Activities in the Neighborhood

The first part is an assessment of direct nature contact activities for children. While the findings showed that children’s lack of exposure to nature was a problem, as shown above, they also emphasized the benefits and importance of natural activities correspondingly [14,15,16]. For example, if children had more time and frequency of natural interactions in the neighborhood, it would increase children’s site identification with the neighborhood environment, benefit their own well-being, stimulate their curiosity and imagination, positively influence their mental health and increase their emotional stability and natural emotional connection. This is why UNICEF declared the well-being of children through nature enjoyment as “the ultimate indicator of a healthy habitat” [17]. Some researchers described that spending time in nature is part of a “balanced diet” of childhood experiences that promote children’s healthy development, well-being and positive environmental attitudes and values [18,19]. The findings are now widely accepted and have helped to spark the interest of government policy makers, educators, urban planners, parents, specialists in child-friendly communities and child-serving volunteers [20,21,22].

### 1.2. Characteristics of Children’s Natural Activities in the Neighborhood

The second part of the topic is on characteristics of children’s outdoor direct experiences of nature. Generally speaking, compared to those of their parents, the frequency and duration of children’s outdoor activities in the neighborhood are significantly decreasing [23,24], and the variability of their locations and modes of activity is evident [25]. There is also a decrease in the scope of children’s neighborhood roaming [26].

In terms of detailed features, children’s natural activities in the neighborhood varied depending on both individual and geographic factors. Previous research in this area had produced mixed findings, but the majority of them suggested that children’s neighborhood natural activities differ depending on their age, gender and geographic location (rural versus urban). For instance, research from the United States [24], Norway [26] and Japan [27] revealed that the frequency of spontaneous natural activities and the range of neighborhood roaming of children would increase with age. For girls, they were subject to more safety concerns from their parents, which is a major barrier to their exposure to natural activities [26]. According to a 10-year follow-up study by the UK government [28], children in the southwest and northeast of England tended to choose the coast as their daily activity site, and the southeast favored green spaces as their neighborhood nature activity site. the proportion of children living in the most deprived situations visit urban greenspace, the coast and the countryside with a lower frequency than children in more affluent areas. The frequency of neighborhood natural activities was significantly higher for rural children than for children in high-density urban communities because rural children have more access to the natural environment [22].

Current related research has focused on developed Western countries, and related studies have concluded significant geographic variability in children’s nature activities [2,7,27,29], which confirms the need to expand the study to more regions, especially in developing countries.

### 1.3. Factors Affecting Children’s Natural Activities in the Neighborhood

The main focus of the third part is the exploration of factors that influence children’s natural activities in the neighborhood. The main reasons for the alienation of children from nature were twofold: the lack of opportunities and the loss of orientation [25]. The former was the well-documented gradual loss of natural space due to factors such as urbanization, which decreased kids’ chances to engage with nature [3,30,31]. For example, due to accelerated urbanization, natural spaces for children to play (fields, riverfront green spaces, and other places for children’s activities) are gradually decreasing. This meant that environmental factors such as regional biodiversity and plant diversity were decreasing, thus reducing the opportunities for children to come into contact with nature on their own [3,30]. Likewise, children’s own excessive academic pressure, parents’ extensive scheduling of interest classes for children outside of school hours [27] and parents’ concerns about the safety of children’s outdoor activities [32] resulted in children not having the time and opportunity to engage in outdoor natural activities. These are also important influencing factors for the lack of opportunities. The latter was the result of factors such as the family environment and the information age, which cause children to become less interested in nature contact activities and lose their orientation to nature contact. For example, the prevalence of electronic media (e.g., TV, computer games and smartphones) caused children to lose appreciation for nature-based activities [27]. The current loss of orientation has focused on the exploration of children’s nature activities from the perspectives of children’s nature connectedness and nature cognition at the environmental psychology level [7,9]. However, despite the existence of certain studies, the interrelationships among predictors have received less attention, and these investigations have been carried out independently by parents without the cooperation of the research subjects (children).

Overall, the previous research mainly focused on urban areas in developed countries, which offers a limited perception of children’s natural activities. As one of the largest developing countries in the world, China has witnessed rapid urbanization over the last decade. Issues arising from urbanization, such as traffic safety and changes in the built urban environment, have made certain impacts on children’s natural activities. It is worth noting that in order to promote the healthy growth of children, the Chinese government has responded positively to the proposal of “child-friendly cities” put forward by the United Nations Children’s Fund. The General Office of the Ministry of Housing and Urban-Rural Development issued the “Guidelines for the Construction of Child-Friendly Spaces in Cities” in 2022, all of which emphasize the importance of children’s community recreational health and natural activities.

Therefore, given China’s specific developmental processes and its culture, in order to understand the characteristics of neighborhood natural activities and their influencing factors among children in China, and to benefit the construction of a current child-friendly city here, we conducted a systematic study of the neighborhood natural activities in terms of frequency, time and location among 900 children in grades 4–5 (ages 10–12) in Lin’an District, Hangzhou City, Zhejiang Province. An attempt was made to explore the following two questions.

(1) Are there differences in children’s neighborhood natural activities in terms of gender and urban-rural areas in Hangzhou, China? The current fertility policy in China has shifted from one-child to multiple-child [33]. In this context, are there differences in neighborhood natural activities between children in one-child and multi-child families?

(2) How do natural activities in children’s neighborhoods relate to children’s nature connectedness and environmental knowledge? Are there mechanisms to enhance nature-based activities in children’s neighborhoods and reduce the risk of “nature deficit disorder” in children?

## 2. Materials and Methods

### 2.1. Ethical Procedures

Ethical approval for the present study was obtained from the Ethics Committee of Zhejiang Agriculture and Forestry University (2022ZAFU0601). The informed written consent of the parents and school principals as well as the verbal consent of the students prior to participation were provided. Additionally, the questionnaire is annotated to indicate that any choice involving privacy can be skipped. This approach meets the criteria already in place for conducting research in a school setting.

### 2.2. Study Area

This study was conducted in Lin’an District, Hangzhou City, Eastern China (Figure 1). Hangzhou is a typical quasi-first-tier city in China, a demonstration area of the commonwealth in Zhejiang Province and one of the key pilot cities for child-friendly cities. Lin’an District is located in the western part of Hangzhou, with an area of 3126.8 km^2^, 5 streets and 13 towns under its jurisdiction. According to the environmental characteristics, Lin’an District is a typical mountainous landscape area. The urban area includes Gongchen Mountain Park, Xijingshan Park, Qingshan Lake Greenway, Maxi Greenway and so on. The rural area includes Campsite Greenway, Baishuijian Scenic Area, Daming Mountain Scenic Area, Qingliangfeng Mountain Park, Taihuyuan Scenic Area, etc. It is noteworthy that there is currently a detailed green space system planning for the urban area, while the rural area only has an overall plan for the spatial development of the countryside. As far as the residents are concerned, in the Lin’an area, there may be significant differences in the quality of recreation between urban areas and rural areas.

According to the Yearbook of Lin’an District People’s Government in 2021, there were 41 primary schools in the district at the end of 2021 (Including 2 private primary schools, with 33,074 students).

### 2.3. Participants

#### 2.3.1. Schools

The primary school affiliated with Zhejiang Agriculture and Forestry University (main office and Ma Xi Campus) and Gaohong Town Center Primary School (Fusheng Campus and Desheng campus) in Lin’an District participated in this study. Both schools are non-gender-specific public schools. ZAFU Affiliated Primary school, in the urban area, has a total of 1828 students. The school has scheduled time for nature education classes every Friday from 3:00 to 4:00 p.m. Nature activities are conducted by professional teachers according to themes such as seasons and animals (Figure 1). There are 1356 students enrolled in Gaohong Town Center Primary School in the rural area. The school makes use of a 200 m^2^ planted field and a 3000 m^2^ mountain forest to conduct nature education classes (Figure 1). The related curriculum is mainly included in the outdoor activity program. In comparison, the nature education programs at both schools are diverse and share many similarities.

It’s also crucial to note that roughly 70% of the students at Gaohong Town Central Primary School are the offspring of migrant workers, some of whose parents do not have the ability to write, which may have an impact on the children’s natural activities, due to their relatively different family backgrounds from children in the ZAFU Affiliated Primary school. The reason will be presented later in the discussion.

#### 2.3.2. Students

Due to the pressure of entrance exams on children in grade 6 and given the relative disadvantage of children in the lower grades (grades 1–3) in terms of topic comprehension, a total of 900 children in grades 4–5 (ages 10–12) from four campuses of the two schools participated in our study (Figure 1). The research focused on children who had lived in the Lin’an area with their parents for more than a year and were able to engage in neighborhood natural activities independently. In total, 855 questionnaires were returned, of which 415 were from ZAFU Affiliated Primary school (157 from the main office and 258 from the Ma Xi campus) and 440 from Gaohong Town Center Primary School (330 from the Fusheng campus and 110 from the Desheng campus). After removing questionnaires with incomplete information, 767 valid questionnaires were collected in total.

### 2.4. Questionnaire Procedure

Before carrying out the survey officially, a pre-survey of 50 questionnaires was conducted by using random sampling on Jincheng Street (Figure 2), Lin’an District, Hangzhou, and the age of the respondents was consistent with the final study, but the relevant data were not involved in the final study analysis. The pre-survey added additional unclear options to the questionnaire, and removed options that were answered less than twice, in order to adjust and optimize the questionnaire to fit the characteristics of the location and natural activities in Lin’an District, Hangzhou.

The formal survey was conducted in May and June 2022, the most suitable time of the year for natural activities in Hangzhou. The optimized questionnaire (Appendix A) was distributed under the supervision of the principal and classroom teachers and was filled out by children directly at school. As suggested by the principal and parents, all questionnaires were in paper format in order to avoid potential distractions from cell phones. Considering the children’s tendency to judge each option to be right or wrong, the children were informed in advance before the questionnaire was distributed that the questionnaire mainly investigated their neighborhood natural activities and their own relationship with nature and that there was no right or wrong for each option. We also designed a small gift (Figure 3) with a small booklet as instant compensation for the questionnaire and promised to conduct one or two nature education activities in the research classes.

### 2.5. Questionnaire Design

The questionnaire (2 pages in Chinese) consists of four main sections: basic information, children’s natural activities in the neighborhood, children’s nature connectedness and children’s environmental knowledge. It takes roughly ten minutes to complete the questionnaire.

#### 2.5.1. Basic Information

The basic information section not only covers the gender, age and place of residence of the children but involves the question “whether you are the only child in your family”. The variables were coded as follows: place of residence (0 = urban, 1 = rural), gender (0 = boy, 1 = girl), and only child status (0 = only child, 1 = multiple children). Age information was used to guarantee the reliability of the questions by removing children whose actual age was not in the range of 10–12 years. Residence information was used to determine the location of children’s home addresses based on the administrative boundaries of urban and rural areas in Lin’an District.

#### 2.5.2. Children’s Natural Activities in the Neighborhood

To understand children’s natural activities in the neighborhood, we investigated in detail the frequency and duration of children’s neighborhood natural activities in the last 30 days [27]. These two questions were differentiated according to 5 levels (see Appendix A for details). The questionnaire also included an open question about natural locations that children frequently visit, which were categorized according to the naturalness of the site (1 = sports field, family yard, community garden, street; 2 = city park, greenway; 3 = forest, field) based on semantic descriptions [34]. For example, in the case of ‘woods’ and ‘forests’, ‘woods’ is classified as a second category of green space for urban development and ‘forests’ is classified as a pure natural space, according to semantic cuts, size and volume, location and artificial components.

#### 2.5.3. Children’s Nature Connectedness

Several techniques are currently available to evaluate natural connectivity [27,35], we chose the natural connectivity metric developed by Brügger [36]. Unlike other techniques that rely on individuals answering abstract items about the relationship with nature, this scale assesses the intimacy between a specific behavior and the natural world in four sections: (a) enjoyment of nature, (b) biological sympathy, (c) belongingness to nature and (d) responsibility for nature. Therefore, this scale is particularly applicable to children [9]. In addition, the scale was adjusted according to the results of the pre-study, for example, the original question “I like to touch plants and animals” was divided into “I like to touch animals” and “I like to touch plants”, because some children in the pre-study indicated that they had differences in their love for plants and animals, so they could not fill in the question. In order to reduce the children’s difficulty in judging the degree, the questionnaire asked the respondents to answer the 15 statements (see Appendix A for details) with “yes” ticked and “no” crossed or not filled in, categorized as “Yes = 1, No = 0” (Cronbach’s alpha = 0.71). When processing data, we adjusted the content of the opposite options (questions 3.3, 3.11, 3.15) and calculated the cumulative value of the final results.

#### 2.5.4. Children’s Environmental Knowledge

To measure children’s natural cognition, Frick and Siegmar Otto’s scale [9,37] was selected and adapted to match the knowledge level of students in grades 4 to 5. The details were adjusted to include common plants and organisms in the Lin’an area, such as replacing the six color photographs of leaves of trees widely distributed in Germany (beech, oak, chestnut, fir, maple, pine) in the original scale with street tree species (cedar, balsam fir) in Hangzhou, China and replacing the German illustration of garbage sorting with a common Chinese style, etc. A total of eight items (see Appendix A for details) were assessed for environmental knowledge, three of which could be answered incorrectly or correctly, and five of which could be answered incorrectly, partially or completely correctly. Questions that were not answered were rated as incorrect. The content of these items reflected a wide range of environmental knowledge, from biology and knowledge of environmental systems to knowledge of environmental actions (Cronbach’s alpha = 0.87). Examples of items are “How many legs does a spider have?” or “Which picture is rice and wheat?” Based on each answer, 1 point for correct, 0.5 points for partially correct, and 0 points for incorrect, with question 5 having two questions for a total of 2 points, we calculated the cumulative score.

### 2.6. Statistical Analysis Plan

A total of 767 valid questionnaires were processed in this study to conduct an analysis of natural activity characteristics and related factors. Data analysis was performed using SPSS version 22.0. For dichotomous variables such as household location (urban or rural), number of children in the household (one child or multiple children) and gender of the child (male or female), we conducted separate independent samples *t*-tests for their children’s neighborhood natural activities to determine whether differences existed across the parameters. Considering some continuous variables had obvious logical relationships with children’s neighborhood natural activities, such as children’s nature connectedness and children’s environmental knowledge, a two-tailed Pearson correlation was used to assess the relationship between children’s neighborhood natural activities and these continuous variables. In addition, the general linear regression model was applied to examine the manner and extent to which the factors affect children’s natural activities in the neighborhood by calculating the standardized regression coefficients of the model.

## 3. Results

### 3.1. Data Description

Ultimately 54.4% of the children who participated in the study were from rural areas and 45.6% were from urban areas. Boys made up 51.2% of the children who answered the questionnaire, while girls made up 48.8%. The percentage of children with only one child in the region (37.1%) was significantly lower than the percentage of multiple children (62.9%).

In terms of the frequency of children’s neighborhood natural activities, approximately 20% of children engaged in less than three neighborhood natural activities per month and 40% of children were reported participating in neighborhood natural activities more than eight times per month. In terms of time spent in children’s neighborhood natural activities, about 75% of children spent between half an hour and two hours in natural activities each time. Regarding the location of children’s neighborhood natural activities, 25.2% of children chose sports fields, home yards, community gardens and streets as activity sites; 46.5% of children chose parks and greenways as recreation sites for natural activities, and 28.3% of children chose pure natural spaces such as forests and fields as natural activity sites (Table 1. for details).

### 3.2. Characteristics of Children’s Natural Activities in the Neighborhood

Between urban and rural areas, the frequency of children’s natural activities was higher in urban areas (M ± SD 3.97 ± 1.12) than in rural areas (M ± SD 3.77 ± 1.24), with significant differences (Figure 4). The naturalness of neighborhood natural activity sites was significantly higher for rural children (M ± SD 2.12 ± 0.71) than for urban children (M ± SD 1.92 ± 0.75). Between the sexes, the duration of natural activity was significantly longer in boys (M ± SD 3.01 ± 1.16) than in girls (M ± SD 2.82 ± 1.0). There is no significant difference in other aspects. Between single and multiple births, singletons were slightly higher in terms of frequency of natural activities, but the difference was not significant. In contrast, the duration of natural activity was significantly higher in singletons (M ± SD 3.05 ± 1.05) than in multiples (M ± SD 2.84 ± 1.15).

### 3.3. Factors Affecting Children’s Natural Activities in the Neighborhood

The correlation results (Table 2) indicated that the frequency of children’s neighborhood natural activities was correlated with the duration of children’s natural activities and children’s nature connectedness. The duration of children’s natural activities in the neighborhood was correlated with children’s neighborhood natural activity locations. The location of children’s neighborhood natural activities was correlated with children’s nature connectedness and environmental knowledge.

The results of the regression models (Table 3) showed that children’s nature connectedness and environmental knowledge significantly and positively influenced the frequency of children’s natural activities. Children’s gender and multiple births significantly predicted the duration of children’s natural activities. Children’s choice of nature activity sites was influenced by the urban and rural environment. In addition, children’s nature connectedness and nature cognition significantly and positively predicted the degree of naturalness of children’s natural activity locations. All three multiple linear regression models above fit significantly.

## 4. Discussion

### 4.1. Characteristics of Children’s Natural Activities in the Neighborhood

The findings show significant differences in children’s neighborhood natural activities between urban and rural areas, with a higher frequency of natural activities among urban children compared with rural children today, which differs from the results in developed countries. Numerous studies [38,39,40,41] have been conducted to prove that children in rural areas have a higher frequency of natural activities than in urban areas. The main reason for this is that the countryside has a better ecological environment in terms of natural texture [40,42]. In contrast to urban children, rural children have direct access to plant diversity, biodiversity and natural outdoor health environments.

It is interesting to note that there is also a small body of research that contradicts this. A study 20 years ago in England and Wales found that, due to restricted access to farmland and a lack of local supervision of children’s activities in rural areas, while having a more natural environment than urban areas, it did not mean that children had more access to nature [43]. A similar study in China 10 years ago found that children in urban schools had less frequent contact with wildlife and plants than children from rural schools, but it was not the urban schools that scored the lowest on nature experiences, but a school in rural Yunnan in Southwestern China, the reason of which was inferred to be left-behind issues [40]. This is a complex issue when viewed as a whole. Compared to the above two studies, our study focuses on children in the Lin’an area of Hangzhou, the result of which is more susceptible to the influence of local urban-rural and regional characteristics. We infer that there are two main reasons why children in urban areas in Hangzhou have a significantly higher frequency of neighborhood natural activities than children in rural areas: (i) Rural children traditionally have abundant natural resources, such as beautiful mountains, grasslands and forests, and have more access to nature, and our study also found that rural children’s nature activities are more natural in rural locations, but at present it does not mean that rural areas have more biodiversity and higher environmental quality than urban areas. For example, relevant studies prove that in today’s industrial transformation, rural industrial companies have higher pollution than those in the city [44,45], while most of the natural environments in China’s cities are systematically planned and designed to be significantly higher in quality than rural areas. Most of the environments in rural China do not meet recreation standards, which may influence the relevant results [3]. (ii) Children’s natural activities are not only affected by the outdoor environment, but also by parental guidance and school education. In the process of our research, we found that 70% of the subjects in the rural areas are migrant families, which require additional communication by telephone because the parents have not mastered the ability to write, and these groups are less educated and less economically capable than those in the urban areas. This may be another reason.

In terms of gender, findings show that boys spend significantly more hours in neighborhood natural activities than girls. Related research suggests that at around 11 years of age, girls are slightly less independently mobile than boys, and girls have a higher proportion of mobility restrictions outdoors than boys, which can affect the way children perform neighborhood natural activities [46,47,48]. In addition, differences in the content of girls’ and boys’ neighborhood outdoor activities may also be a factor, such as boys’ preference for outdoor natural activities, including dragonfly catching, research, water treading, etc. [49]. At present, there is no consensus on gender differences in research, and more sample interventions are needed.

In terms of single-child and multiple-child family structures, the findings suggest differences in the duration of children’s neighborhood natural activities. Objectively, there may be differences in the types of natural activities between single-birth and multiple-births due to certain differences in their upbringing, with singleton children being more independent and non-singleton children being more interactive in natural activities. Due to the number of factors involved, including the way of participating in natural activities (active or parent-led participation), multiple-child family characteristics and parenting differences, at present, there is no difference in the frequency and location of natural activities, and the differences in the timing of nature-based activities cannot be fully explained at this time. The effects of family structure have not yet been the subject of investigations. In the future, the performance of children’s natural activities in terms of family structure features will be a crucial subject in the context of China’s multiple-child policy.

### 4.2. Impact of Nature Connectedness and Children’s Environmental Knowledge

Our results show that children’s nature connectedness significantly and positively predicts the frequency of children’s natural activities. The findings of this section of the study support previous studies that increasing children’s nature connectedness increases the frequency of children’s natural activities [7,27].

At the same time, our study concluded that children with higher levels of nature connectedness would be more likely to choose sites with higher levels of naturalness as their frequent locations for natural activities. We believe that this is a mutually positive influence, with children who have more exposure to purely natural spaces having increased nature connectedness during re-experience natural activities [7], and their nature connectedness would also promote children to choose more naturalistic locations to experience.

Our findings demonstrate that environmental knowledge significantly and positively predicts the frequency of children’s neighborhood natural activities and the location of nature neighborhood nature activities, which complements the value of environmental education in terms of nature neighborhood natural activities. The central reason for this is that children who have more knowledge about the environment will have higher performance in terms of desire for nature exploration and interest in the experience. In this regard, education in the school classroom, home education by parents and participation in nature education activities outside the classroom are all excellent ways to enhance children’s environmental knowledge and reduce the risk of nature-deficit disorder in children.

Our study does not find a relationship between nature connectedness, environmental knowledge and children’s neighborhood natural activity hours, which is in line with other studies that have shown that activity hours are influenced by green space quality, biodiversity and children’s activity patterns at activity sites [50,51].

### 4.3. Limitations

There are two main limitations to our study: (i) The impact of COVID-19. Although there have been no confirmed cases of COVID-19 in Hangzhou for twelve months, perhaps the frequency and duration of children’s natural activities are implicitly influenced by epidemic prevention policies and parents worried about the impact of an outbreak [32]. (ii) Limitations of the child population. In rural areas, informed consent forms were given to the parents of the children studied. Parents from rural areas gave significantly less permission than in urban areas for the children to participate in the questionnaires. There, the number of children whose parents cannot write and read is limited.

## 5. Conclusions

Due to the many benefits of nature contact, children’s natural activities in the neighborhood, a form of direct contact with nature, have been widely investigated in urban areas of developed countries. In order to figure out the characteristics and some of the influencing factors of children’s neighborhood natural activities in China, and to benefit the construction of the current child-friendly city, this study conducted a systematic questionnaire survey of 900 children (10–12 years old from Hangzhou City, Zhejiang Province, Eastern China) about their natural activities in the neighborhood. By using separate independent samples t-tests, Pearson correlation and general linear regression model, the study showed significant variability in children’s natural activities in Hangzhou, China, in terms of location, gender and multiple births. The results also suggested that increasing children’s nature connectedness and environmental knowledge can increase the frequency of children’s neighborhood natural activities and the selection of pure nature spaces. This gives policy makers a new way to promote children’s neighborhood natural activities by educating children about nature and nurturing nature connections. In the future, it would be a good topic to examine what types of policy measures and educational programs can achieve such effects.

In general, children’s natural activities in the neighborhood involve many dimensions, including the influence of urban planning, green space layout, family environment, family education, academic pressure electronic media and many other dimensions. Last but not least, more regions and samples are needed to enrich such studies in the future.

## Figures and Tables

**Figure 1 ijerph-19-16087-f001:**
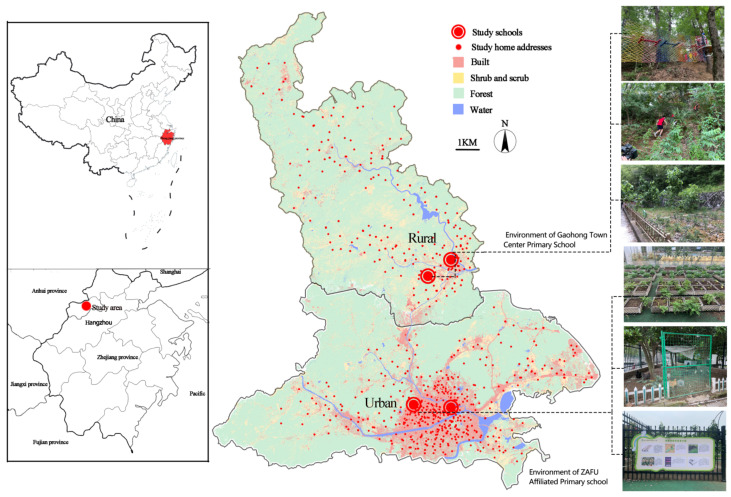
Research area environment.

**Figure 2 ijerph-19-16087-f002:**
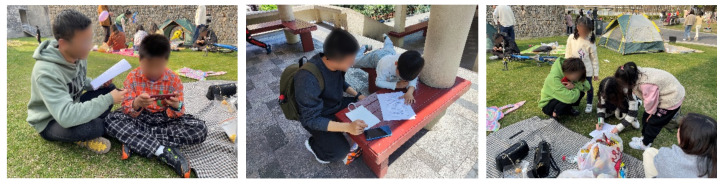
Picture of pre-survey.

**Figure 3 ijerph-19-16087-f003:**
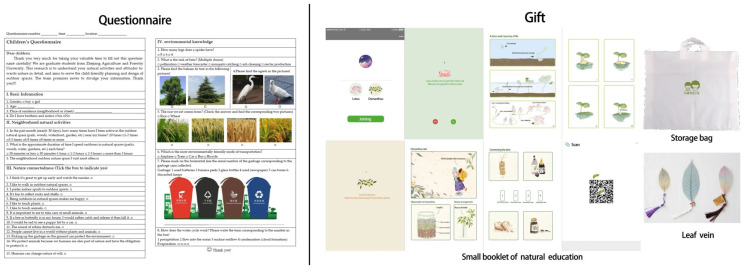
Questionnaire and gift.

**Figure 4 ijerph-19-16087-f004:**
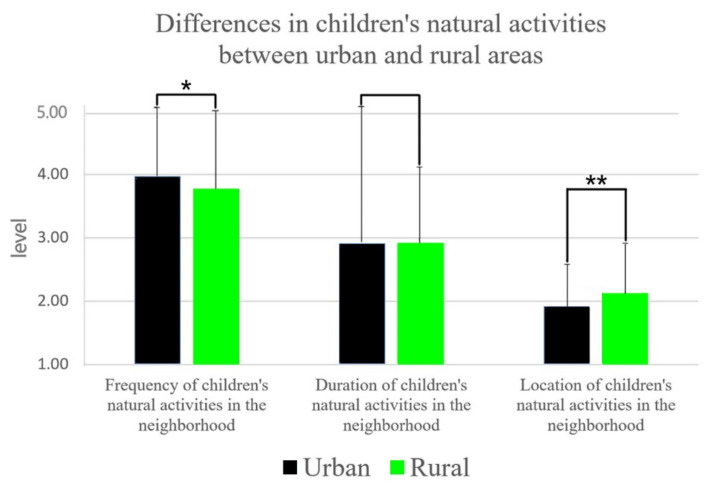
Differential description of children’s natural activities in the neighborhood; *n* = 26, mean ± SE, * *p* < 0.05, ** *p* < 0.01 determined by the paired *t*-test.

**Table 1 ijerph-19-16087-t001:** Descriptive statistics for relevant variables.

Variable	Mean	N	SE	Median	Range
Frequency of children’s natural activities in the neighborhood	3.85	767	0.04	4	1–5
Duration of children’s natural activities in the neighborhood	3	767	0.07	3	1–5
Location of children’s natural activities in the neighborhood	2	767	0.03	2	1–3
Children’s nature connectedness	11	767	0.04	11	1–15
Children’s environmental knowledge	6.58	767	0.05	6	1–9

**Table 2 ijerph-19-16087-t002:** Two-tailed Pearson correlation of three children’s natural activity parameters with each other and with children’s nature connectedness and environmental knowledge.

		Frequency of Children’s Natural Activities in the Neighborhood	Duration of Children’s Natural Activities in the Neighborhood	Location of Children’s Natural Activities in the Neighborhood	Children’s Nature Connectedness	Children’s Environmental Knowledge
Frequency of children’s natural activities in the neighborhood	Pearson Correlation	1				
	Sig. (two-tailed)					
Duration of children’s natural activities in the neighborhood	Pearson Correlation	0.198 **	1			
	Sig. (two-tailed)	0.000				
Location of children’s natural activities in the neighborhood	Pearson Correlation	−0.037	0.097 *	1		
	Sig. (two-tailed)	0.360	0.017			
Children’s nature connectedness	Pearson Correlation	0.152 **	0.015	0.099 *	1	
	Sig. (two-tailed)	0.000	0.692	0.016		
Children’s environmental knowledge	Pearson Correlation	0.030	0.072	0.163 **	0.161 **	1
	Sig. (two-tailed)	0.421	0.052	0.000	0.000	

* *p* < 0.05, ** *p* < 0.01.

**Table 3 ijerph-19-16087-t003:** Standardized regression coefficients for factors to predict three children’s natural activity parameters and the fitness of the model R^2^ and F.

	Residential Area	Gender	Family Fertility Status	Children’s Nature Connectedness	Children’s Environmental Knowledge	R^2^	F
Frequency of children’s natural activities in the neighborhood	0.04	0.04	−0.04	0.01 *	0.14 **	0.03	4.13 **
Duration of children’s natural activities in the neighborhood	−0.01	0.08 *	−0.08 *	0.07	0.01	0.05	5.25 **
Location of children’s natural activities in the neighborhood	−0.17 **	0.03	0.01	0.14 **	0.11 **	0.07	7.16 **

* *p* < 0.05, ** *p* < 0.01.

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
