# Peer review of "A Study on the Characteristics of Children’s Natural Activities in the Neighborhood and Their Influencing Factors: Evidence from Hangzhou, China"

_ijerph, 2022, doi:10.3390/ijerph192316087_

Round 1
Reviewer 1 Report
This is a very interesting study, very dilligently worked-out.. spending lots of time working on it and the results are interesting. Here are a few points that should be either improved or considered to be improved:
l. 123: check the first use of „the study“ in the text „the study conducted a systematic study
l. 132: preposition missing: How do natural activities in children's neighborhoods relate to children's nature of? connectedness and nature awareness
l. 157: not capital T: there were...
l. 178: check the comma and space : have the ability to write,which may
l. 180 – I suggest to substitue the word „weakness“(it is a normal development feature, not a weakness) – in the text „the relative weakness of children in the lower grades (grades 1-3) in terms of topic comprehension“
l. 182: not capital A: a total of 900 children...
l. 209: I cannot imagine what is „combing leaf veins“ – I cannot see it in the picture
l. 356 – capital A: this. A study 20 years...
l. 406: - add apostroph after children: children’ nature connectednes
l. 417: you say that „Our findings demonstrate that environmental knowledge significantly and positively predicts the frequency of children's neighborhood natural activities ..“ and then in l.427 you are say that „Our study did not find a relationship between nature connectedness,,,, and children's neighborhood natural activity...“ What is the good statement?
You say (L. 178) that some parents were migrant workers not able to write,which may have an impact on the children's natural activities – why this suggestion? what kind of impact? Explain´
l. 428 – I suggest to substitue the expression „children's neighborhood natural activity“ here (l. 428 and in the whole study) with the better sounding word group „children's natural activities in the neighborhood“
I can see in all the pictures about the survey process, that there was an adult helping the child. Was this a rule? Or just with some of the children? Has the presence of the adult not influenced the filling of the questionnaire and distort the answers with some/with all/in the overall sample?
Author Response
Dear Reviewer
please forgive my carelessness (my last response was misplaced), this pdf is my response to your valuable comments.
Kind regards,
Rui Ji
Zhejiang Agriculture and Forestry University
2022 11.25

Reviewer 2 Report
The article is devoted to the study of the connection of children with nature and the factors that influence this connection. This study is of undoubted relevance, as it provides information about children's knowledge of the world around them and their behavior.
The text presents the result of a sociological study. The authors correctly presented the methods and results. In the discussion of the results, the author's understanding of the current state of affairs is presented.
However, the article needs to be improved.
1. The abstract does not provide a clear statement of purpose. The purpose is guessed from the context, but the reader must adequately understand the author's position.
2. The authors link the problem of children being isolated from nature with urbanization (page 1). This is an indisputable fact. However, this is not the only factor that separates children from the environment. The authors need to touch upon the issues of using gadgets, the Internet, the level of criminal problems, etc. Also important in the Introduction is the issue of children's employment. Increasing demands on children require their parents to take them to additional classes in sports sections, robotic schools for children, art, etc. Children have less time for walks and learning about the world around them and communicating with each other. And this is an objective fact.
The problem raised by the authors is very acute, although it is hidden behind the mask of well-being. Therefore, it is necessary to make sharper in the Introduction.
3. The conclusion needs to be expanded. The authors should present the goal, what methods were used to solve the set research tasks, and briefly outline the main results. And after that - the text, which is now the Conclusion. In this case, the author's conclusion will be weighty and justified.
Author Response
Dear Reviewer
Thank you for your valuable comments concerning our manuscript " ijerph-2055727". This paper has been revised point-by-point with your comments. The details are as follows:
Review Comments 1:
The abstract does not provide a clear statement of purpose. The purpose is guessed from the context, but the reader must adequately understand the author's position.
The author's answer: Thank you for your suggestion, the paper has been supplemented with the purpose of the study in the abstract.
Review Comments 2:
The authors link the problem of children being isolated from nature with urbanization (page 1). This is an indisputable fact. However, this is not the only factor that separates children from the environment. The authors need to touch upon the issues of using gadgets, the Internet, the level of criminal problems, etc. Also important in the Introduction is the issue of children's employment. Increasing demands on children require their parents to take them to additional classes in sports sections, robotic schools for children, art, etc. Children have less time for walks and learning about the world around them and communicating with each other. And this is an objective fact.
The problem raised by the authors is very acute, although it is hidden behind the mask of well-being. Therefore, it is necessary to make sharper in the Introduction.
The author's answer: Thank you very much, this is a very good suggestion. This section has been added. On the one hand, we have added them in section 1.3 Factors affecting children's natural activities in the neighborhood. These reasons are divided into two parts: the lack of opportunities and the loss of orientation (Red font is additional content).
The former was the well-documented gradual loss of natural space due to factors such as urbanization, which decreased kids' chances to engage with nature. For example, due to accelerated urbanization, natural spaces for children to play (fields, riverfront green spaces, and other places for children's activities) were gradually decreasing. This meant that environmental factors such as regional biodiversity and plant diversity were decreasing, reducing the opportunities for children to come into contact with nature on their own. Likewise, children's own excessive academic pressure, parents' extensive scheduling of interest classes for children outside of school hours, and parents' concerns about the safety of children's outdoor activities resulted in children not having the time and opportunity to engage in outdoor natural activities, which are also important influencing factors for the lack of opportunities. The latter was the result of factors such as the family environment and the information age that cause children to become less interested in nature contact activities and lose their orientation to nature contact. For example, the prevalence of electronic media (e.g., TV, computer games, and smartphones) caused children to lose appreciation for nature-based activities.
One the other hand, they are also mentioned in the final outlook section and are one of the elements that will be very worthy of study in the future.
Review Comments 3:
The conclusion needs to be expanded. The authors should present the goal, what methods were used to solve the set research tasks, and briefly outline the main results. And after that - the text, which is now the Conclusion. In this case, the author's conclusion will be weighty and justified.
The author's answer: Thank you very much for your careful review of this article. In conjunction with your comments on the conclusion section, the purpose and methods of the study have been added to the conclusion section of this paper. See the text for details l.456-l. 464.Specific additions are listed below:
In order to figure out the characteristics and some of the influencing factors of chil-dren's neighborhood natural activities in China, and to benefit the construction of the current child-friendly city, this study conducted a systematic questionnaire survey of 900 children (10-12 years old from Hangzhou City, Zhejiang Province, eastern China) about their natural activities in the neighborhood. By using separate independent samples t-tests, Pearson correlation and general linear regression model,the study showed ······
Special thanks to you for your good comments.
Kind regards,
Rui Ji
Zhejiang Agriculture and Forestry University
2022 11.25